# Short-Term Effects of Centralization of the Glenohumeral Joint and Dynamic Humeral Centering on Shoulder Pain, Disability, and Grip Strength in Patients with Secondary Subacromial Impingement Syndrome

**DOI:** 10.3390/healthcare11222914

**Published:** 2023-11-07

**Authors:** Yong-Hee Kim, Hwi-Young Cho, Sung-Hyeon Kim

**Affiliations:** 1Department of Health Science, Gachon University Graduate School, Incheon 21936, Republic of Korea; yonghee1018@gmail.com; 2Department of Physical Therapy, College of Health Science, Gachon University, Incheon 21936, Republic of Korea

**Keywords:** centralization of glenohumeral joint, dynamic humeral centering, subacromial impingement syndrome

## Abstract

Subacromial impingement syndrome (SIS) is one of the most common causes of shoulder pain in adults and is caused by muscle imbalance around the shoulder joint, which is referred to as secondary SIS. Centralization of the glenohumeral joint (CGH), one of the intervention methods for this, targets strengthening the control ability of the rotator cuff. Dynamic humeral centering (DHC) targets the learning of selective contractile function of the pectoralis major and latissimus dorsi as depressors of the humeral head. This study aims to determine the short-term effects of CGH and DHC on pain, disability, and grip strength in patients with secondary SIS. Forty-eight patients with secondary SIS participated in the study and were randomly allocated into three groups (CGH group (*n* = 16), DHC group (*n* = 16), and simple exercise group (*n* = 16)) and received the intervention for 50 min. The Constant–Murley score was used to assess shoulder pain and disability (primary outcome), and a hand-held dynamometer was used to assess grip strength (secondary outcome). Measurements were performed before the intervention and one day after the intervention. The results showed that the Constant–Murley score improved in the CGH and DHC groups. In addition, pain and disability (range of motion scores) improved in both the CGH and DHC groups. Improvements in disability (shoulder strength) and grip strength were seen only in the CGH group. Both CGH and DHC can be used as methods for short-term pain release and disability recovery in secondary SIS. In particular, CGH appears to be more effective in the short-term improvement in shoulder strength and grip strength.

## 1. Introduction

Shoulder complaints are a medically and socioeconomically significant problem [1]. Shoulder complaints are reported to be highly prevalent in adults [2,3,4]. Subacromial impingement syndrome (SIS) is one of the most common causes of adult shoulder complaints [5]. In Korea, rotator cuff syndrome and SIS are the leading causes of shoulder disease [6]. SIS causes pain and disability at work and during daily activities [7,8].

SIS is a disease in which impingement occurs between the rotator cuff and the coracoacromial arch in the subacromial space [9]. SIS is divided into primary and secondary impingement [9,10,11,12,13,14]. Primary impingement is caused by a structural narrowing of the subacromial space due to inflamed tissue or bony growth. Secondary impingement is mainly caused by a functional disturbance in the centering of the humeral head due to muscle imbalance, such as rotator cuff weakness. Specifically, localized functional disorders occur more often during secondary rather than primary impingement [12,13,14,15,16,17,18,19]. Various causes of the shoulder girdle complex instability, such as scapular dyskinesis, insufficient scapular motor control, and rotator cuff pathology, can lead to secondary SIS. Secondary impingement changes the biomechanics of the shoulder girdle, and it can lead to abnormal stresses in the anterior capsular structure, increased possibility of compression of the rotator cuff, and decreased performance [20]. In addition, shoulder injuries can impair sensory-motor control and limit force transfer to the upper extremity, causing physical disability or functional impairment related to the performance of activities of daily living (ADLs) or negatively impacting hand function [21,22]. Patients suffering from SIS often experience decreased handgrip strength, and grip strength can be used as an indicator to monitor rotator cuff function [23,24]. The function of the upper extremities is essential for most activities of daily living. Therefore, the restoration of shoulder stability and disability, as well as pain control, is considered very important for the rehabilitation of shoulder injuries in clinical practice [25].

Several conservative modalities, such as electrotherapy, lasers, and ultrasonic waves, have been proposed to improve SIS [26,27,28,29,30]. However, there is limited evidence regarding the efficacy of these therapeutic interventions. Preventing the impingement of the coracoacromial arch by positioning the humeral head in the glenoid fossa during movement can alleviate SIS symptoms [31,32,33,34]. Shoulder stability refers to the condition where the humeral head swiftly aligns correctly within the glenoid fossa [35]. The glenohumeral joint can maintain alignment through control by the rotator cuff or downward movement by the pectoralis major and latissimus dorsi. For this purpose, the centralization of the glenohumeral joint (CGH) and dynamic humeral centering (DHC) are interventions used to improve SIS symptoms [36,37]. CGH is an intervention to enhance rotator cuff control and it adjusts the positioning of the humeral head to the center of the glenohumeral joint. DHC is a method of inducing movements that depress the humeral head [38] and that improve the co-contraction function of the pectoralis major and latissimus dorsi as depressors of the humeral head. These interventions can reduce pain and improve shoulder function in patients with SIS. However, among the therapeutic interventions for SIS, interventions to stabilize the glenohumeral joint have yet to be studied. Furthermore, although CGH and DHC employ different approaches for ameliorating SIS, there is a need to examine the efficacy of these two treatments more closely.

This study was designed to determine the short-term effects of CGH and DHC on secondary SIS symptoms and provide evidence of therapeutic efficacy. This study aimed to determine the short-term effects of CGH and DHC on shoulder pain, disability, and grip strength in patients with secondary SIS.

## 2. Materials and Methods

### 2.1. Participants

The participants enrolled in this study were patients with shoulder pain. All volunteers were considered for inclusion by the same evaluator. The inclusion criteria for this study were (1) diagnosed with secondary impingement, (2) age greater than 30 years, (3) positive in two or more of the three impingement syndrome tests (Neer test, Hawkins test, and Yocum test), (4) shoulder pain duration longer than one month, and (5) a Constant–Murley score < 80. The study exclusion criteria included patients presenting with specific shoulder conditions such as (1) reduced passive range of motion, (2) tendinous calcification in shoulder joint, (3) corticosteroid injection within the previous 30 days, (4) neurological pain caused by compression of the cervical vertebra, (5) dislocation of the glenohumeral joint, (6) history of shoulder surgery, (7) history of fracture in clavicle, scapular, and humerus, and (8) inflammatory or neoplastic disease. All participants received a full explanation (process of the study, risk, and the possibility of side effects) about this study before signing a consent form. This study was approved by the Institutional Review Board of Gachon University (1044396-201910-HR-185-01), enrolled in the Clinical Research Information Service that complies with the World Health Organization International Clinical Trials Registry Platform (registration number: KCT0006025). All study procedures were performed in compliance with the Declaration of Helsinki.

The sample size was determined via the G-power software (version 3.1.9.4; Heinrich Heine University, Dusseldorf, Germany) [39]. We established an effect size f of 0.25, an alpha level of 0.05, and a power of 0.8 [40]. The required sample size for the three groups was calculated to be 42 participants, which became 46 participants after accounting for a 10% dropout rate during the study. But, to equalize the number of subjects between groups, a total of 48 subjects were finally calculated.

### 2.2. Study Design and Protocol

The study was a single-center randomized controlled trial with blinded patients and an assessor. Random assignment took place after the anthropometric measure, questionnaire, and baseline assessments. Randomization of the three treatments (CGH, DHC, and simple exercise (SE)) was in a 1:1:1 ratio and was performed using stratified permuted block randomization (with a block size of four). Stratification factors were sex (male/female), age (30 s/40 s/50 s), and affected side (left/right). Outcome measurements were performed by an evaluator who was blind to group allocation. Intervention and evaluation were performed on separate days for the groups to prevent information on stretching and exercise methods from being transmitted between groups. The same researcher provided all interventions. Figure 1 is a flowchart depicting the experimental procedure of this study. 

### 2.3. Intervention

In this study, the intervention was administered by an experienced physiotherapist with a specialist manual therapy certification and at least six years of clinical experience in shoulder treatment. All groups received a 50 min intervention consisting of a warm-up (10 min), main exercise (30 min), and cool-down (10 min) (Table 1). The warm-up consisted of stretching, massaging the muscles and soft tissues of the neck and shoulders, and scapular stabilization. The main exercises were CGH exercise, DHC exercise, and simple exercise for each group (Figure 2). The cool-down consisted of stretching the muscles and soft tissues of the neck and shoulders.

#### 2.3.1. Centralization of the Glenohumeral Joint Exercise

CGH consisted of passive movement by the physical therapist and self-exercise moving against resistance (Table 1, Figure 2A–D). CGH is an intervention that involves positioning the humeral head in the center of the glenohumeral joint [36,37,41]. The participant learned to passively move the humeral head in four directions (superior, inferior, right, and left). The participants were then trained to position the humeral head in the center of the glenohumeral joint by holding the band with one hand on a cushioned ball or against the resistance exerted by the physical therapist.

#### 2.3.2. Dynamic Humeral Centering Exercise

The DHC exercise method is presented in Table 1 and Figure 2A,B,E,F. DHC is a training method that depresses the humeral head while moving the upper extremity [38,42]. The participant first learned to push the humeral head during passive shoulder abduction. The participant was then trained to co-contract the pectoralis major and latissimus dorsi during active shoulder abduction.

#### 2.3.3. Simple Exercise

Participants received active and passive range of motion (ROM) exercises within the pain-free range and traditional exercises such as pendulum exercises. Active and passive exercises were performed in the degrees of freedom of the shoulder (flexion, extension, abduction, adduction, internal rotation, external rotation, horizontal abduction, and horizontal adduction). The pendulum movement was performed in the vertical direction, horizontal direction, and circumduction (clockwise and counterclockwise) (Table 1, Figure 2A,B,G).

### 2.4. Outcome Measurements

The primary outcome (pain and disability) and secondary outcome (grip strength) were set to evaluate the effectiveness of the intervention. The Constant–Murley score was used to assess the primary outcome, and a hand-held dynamometer was used to evaluate the secondary outcome. Assessments were conducted before the intervention and one day after the intervention at the same time of day as before the intervention. Two physical therapists with at least three years of clinical experience and a master’s degree performed the blinded evaluations. All evaluations were carried out in an isolated, well-lit room with a temperature ranging from 23 to 24 °C, devoid of external interference.

#### 2.4.1. Constant–Murley Score

The Constant–Murley score is an assessment tool for assessing pain and disability in patients with shoulder disease and it shows excellent reliability (ICC = 0.93–0.95) [43]. The Constant–Murley score evaluates overall disability using subjective and objective measurements of the shoulder of the patient. The components of the Constant–Murley score include pain and disability (ROM, strength, ability to perform ADLs, and upper extremity function). The test consists of four subscales (pain = 15 points, ADLs = 20 points, ROM = 40 points, and strength = 25 points) for a total score of 100 points. The maximum pain score is 15, and the degree of current shoulder pain is indicated on a 15 cm line. ADLs have a maximum score of 20 and are evaluated daily. Movement has a maximum score of 40 and is assessed based on the current ROM of the shoulder joint. Strength was estimated as the weight (in pounds) that the subject could withstand while maintaining a 90° abduction posture in the scapular plane. The maximum possible score is 25.

#### 2.4.2. Grip Strength

Grip strength was measured using a hand-held dynamometer (Jamar digital hand dynamometry, Jamar Instruments, Chicago, IL, USA) that showed excellent reliability (ICC = 0.94–0.98) [44,45]. The participants performed the grip strength measurements while being seated in a chair without armrests. They were instructed to keep the humerus attached to the trunk, the elbow joint flexed at 90°, and the forearm in a neutral position. The participants were instructed to assume a comfortable position for gripping their wrists (0–30° extended, 0–15° tilted toward the wrong side) and to grasp the grip dynamometer as strongly as possible. The highest value among the three repeated measurements was used for grip strength, and the results were expressed in pounds (lb).

### 2.5. Data Analysis

Statistical analysis was performed using the SPSS statistical software (version 25.0; SPSS Inc., Chicago, IL, USA). The Kolmogorov–Smirnov test confirmed the normality of all variables. All continuous variables were expressed as the mean ± standard deviation, and nominal variables were presented as a ratio. The general characteristics and baseline of the three groups were compared using a one-way ANOVA test with a post hoc Tukey HSD test and Chi-square test. Repeated measures ANOVA was used to analyze changes between groups over time. Sphericity was confirmed using Mauchly’s sphericity test, and violations of sphericity were determined using the Greenhouse–Geisser estimate of degrees of freedom. Levene’s test confirmed the homogeneity of variances. Post hoc analysis was performed using a paired *t*-test and one-way ANOVA using Tukey’s HSD test. A *p*-value of <0.05 was considered statistically significant.

## 3. Results

### 3.1. Demographic and Clinical Profiles of the Study Participants

Forty-eight participants were recruited for this study and completed all procedures without dropping out. There were no significant differences between groups in the general characteristics of the study participants (Table 2). All participants showed positive reactions in at least two impingement tests, with no significant difference in the positivity rate for each test among the groups. 

### 3.2. Constant–Murley Score

The Constant–Murley scores are shown in Table 3 and they increased significantly in the CGH and DHC groups after treatment compared to before treatment (*p* < 0.05). These scores were significantly higher in the CGH group compared to those in the SE group after treatment (*p* < 0.05). Among the detailed items of the Constant–Murley score, pain increased significantly after treatment compared to that before treatment in the CGH and DHC groups (*p* < 0.05), and there was no significant difference between the groups.

There were no significant changes in ADLs. ROM values increased significantly after treatment compared to those measured before treatment in all groups (*p* < 0.05), and the CGH group had significantly higher ROM values than the SE group after treatment (*p* < 0.05). Shoulder strength increased significantly after treatment compared to that before treatment in the CGH group only (*p* < 0.05).

### 3.3. Grip Strength

The grip strength values are listed in Table 3. Grip strength showed a significant increase after treatment compared to that before treatment in the CGH group only (*p* < 0.05).

## 4. Discussion

We aimed to determine the short-term effects of CGH and DHC interventions on shoulder pain, disability, and grip strength in patients with secondary SIS. We drew the following two conclusions from our data. First, the Constant–Murley scores increased in the CGH and DHC groups. Among the detailed items, pain and ROM were improved in both the CGH and DHC groups, whereas shoulder strength was only enhanced in the CGH group. Second, the CGH group showed an improvement in grip strength.

CGH involves training for the active centering of the humeral head that has deviated from the glenoid fossa by controlling it with the rotator cuff [36,37,41]. CGH is a method of maintaining the humeral head in a neutral position by using resistance and body weight in various postures. DHC is a training exercise that induces depression of the humeral head through the selective contraction of the pectoralis major and latissimus dorsi muscles while actively abducting the arm [46,47]. DHC is a muscle contraction method that alleviates the impingement caused by degenerative rotator cuff disease. CGH and DHC treatments can inhibit the elevation of the humeral head during abduction in patients with SIS, resulting in increased mobility and amelioration of pain. Our data show that a single intervention improved the total Constant–Murley score, indicating short-term improvement in shoulder disability. Detailed items such as pain (improvement of 26.94% for CGH and 16.97% for DHC) and ROM (improvement of 23.62% for CGH and 21.56% for DHC) also showed improvement. These results are consistent with those of previous studies and demonstrate that CGH and DHC are effective in improving joint stability and reducing pain [31,37,38]. However, no improvement was observed in ADLs assessments. These changes were validated one day after treatment. However, it appears that a one-time intervention is insufficient to affect dynamic and occupational activities, such as ADLs.

Movement in the scapular plane is a posture where the distance between the coracoacromial arch and the humeral head is maintained more widely than in the frontal plane [48]. When moving at the same angle, abduction in the scapular plane has a lower possibility of impingement than in the frontal plane. It requires less depressed movement of the humeral head. The DHC trains the participants to induce early activation of the pectoralis major and latissimus dorsi muscles during abduction in the scapular plane. However, DHC is a method of learning early contraction of the pectoralis major and latissimus dorsi as depressors of the humeral head rather than maintaining a static posture or improving upward force [46,47]. This explains why there was no change in shoulder strength in the DHC group. Shoulder strength was assessed by employing the Constant–Murley protocol to ascertain the resistance in a fixed position (90° abduction in the scapular plane) [43]. Therefore, the short-term effects of DHC that involve training in a dynamic scenario rather than in a static posture did not seem to have an effect. On the other hand, CGH is training that improves resistance in all directions through body weight and external resistance [36,37,41]. Thus, the isometric ability of the muscles around the shoulder, including the rotator cuff, can be improved [49,50]. As the isometric contraction method of the shoulder muscles performed in the CGH is similar to that used in shoulder strength evaluation, the effect of short-term training appears to have been realized.

Hand function is related to the overall function of the upper extremities, and a decline in upper extremity function is associated with disability, which is the inability to perform ADLs properly. The CGH used in this study consisted of training to keep the humeral head at the center of the glenoid fossa in various shoulder postures [21,22]. Through this training, the ability of the rotator cuff and surrounding muscles to control the humeral head can be improved. Increased shoulder joint stability due to enhanced control of the rotator cuff may also affect hand function [21,22,23,24]. Our results showed an improvement in grip strength in the CGH group. These results show that the function of the rotator cuff improved in the short term through CGH. In the DHC group, grip strength did not improve due to the characteristics of the training method, as discussed above. Functions enhanced by DHC are related to movements that involve depressive movements of the humeral head, such as abduction of the shoulder. However, grip strength is an assessment that does not include shoulder movement. As a result, even if short-term shoulder pain or functional changes occur due to DHC, they do not affect hand function.

This study investigated the effects on shoulder function following interventions applied to the glenohumeral joint. However, there are some limitations. First, the small sample size restricts the generalizability of the study’s findings. Second, since the post-evaluation was conducted only once, it is impossible to know whether the effect of the intervention is temporary or lasting. Third, this study confirmed only the short-term effects of exercise intervention. Therefore, it was impossible to know the effectiveness of long-term exercise treatment or the duration of the effect. However, by identifying short-term effects, it was possible to obtain evidence on which to base long-term treatment. Finally, the anatomical changes could not be identified because only functional assessments were conducted. Future studies should examine the effects of CGH and DHC on pain, disability, and grip strength in secondary SIS. First, the duration over which a single session positively affects symptoms should be determined. This will allow us to determine the duration between sessions of these interventions. Second, the difference in effectiveness between a single session and multiple sessions will need to be determined. Third, CGH and DHC’s effects on other symptoms accompanying secondary SIS must be determined.

## 5. Conclusions

This study demonstrated that CGH and DHC interventions reduced shoulder pain and improved disability in patients with secondary SIS. CGH, an exercise that enhances the isometric contraction capability of the rotator cuff muscles, effectively increases shoulder strength and grip strength despite being a single intervention. Both therapeutic interventions effectively relieved shoulder pain and reversed shoulder disability, but CGH was more effective in improving hand function.

This is the first study to validate the effectiveness of CGH and DHC as interventions used in clinical practice to intervene in pain and disability caused by secondary SIS, and we hope that many physical therapists will utilize them in clinical practice based on our results. Of course, a significant limitation of this study is that the intervention was a single session, and the effects of multiple sessions or long-term effects need to be clarified. Therefore, further research on CGH and DHC for secondary SIS is needed.

## Figures and Tables

**Figure 1 healthcare-11-02914-f001:**
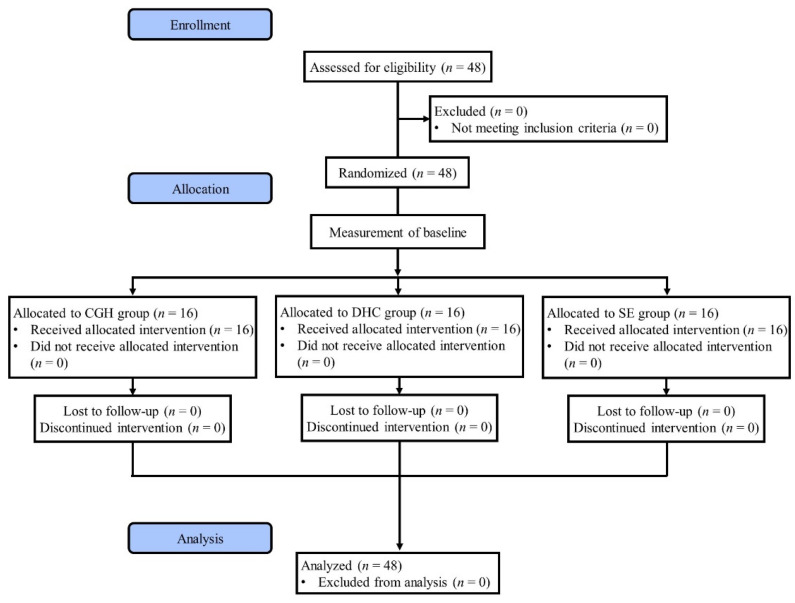
Flowchart of the study.

**Figure 2 healthcare-11-02914-f002:**
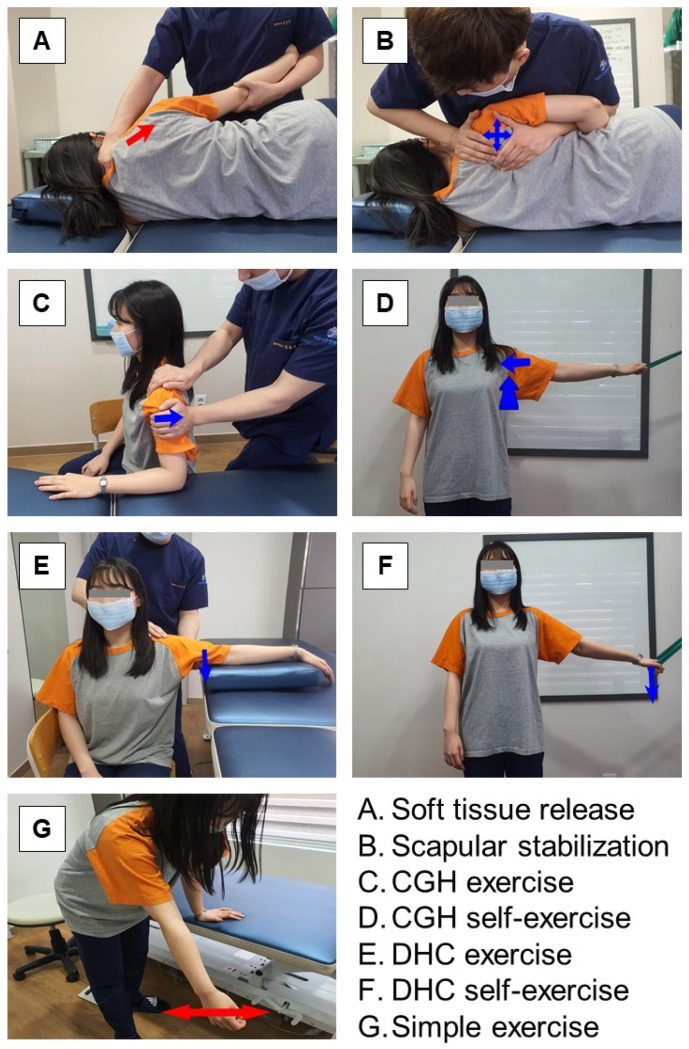
Exercise intervention for subacromial impingement syndrome. CGH, centralization of the glenohumeral joint. DHC, dynamic humeral centering.

**Table 1 healthcare-11-02914-t001:** Description of the intervention.

	CGH	DHC	SE	Time
Warm-up	Step 1: Stretching and massage of muscle and soft tissue of the neck and shoulder region Step 2: Scapular stabilization	Step 1: Stretching and massage of muscle and soft tissue of the neck and shoulder region Step 2: Scapular stabilization	Step 1: Stretching and massage of muscle and soft tissue of the neck and shoulder region Step 2: Scapular stabilization	10 min
Main exercise	Step 1: Passive movement of the humeral head in four directions by the therapist Step 2: The participant holds a band or cushion ball, and, against resistance, positions the head of the humerus in the center of the glenohumeral joint	Step 1: Lowering of the humeral head during the passive abduction of the shoulder Step 2: Actively lowering the humeral head by co-contraction of the pectoralis major and latissimus dorsi during active abduction of the shoulder	Step 1: Active and passive range of motion exercise in the pain-free range of motion (flexion, extension, abduction, horizontal abduction, horizontal adduction, external rotation, and internal rotation) Step 2: Movement and pendulum exercise (anterior–posterior, medial–lateral, and rotation)	30 min
Cool-down	Stretching Muscle and soft tissue of the neck and shoulder region	Stretching Muscle and soft tissue of the neck and shoulder region	Stretching Muscle and soft tissue of the neck and shoulder region	10 min

CGH, centralization of the glenohumeral joint. DHC, dynamic humeral centering. SE, simple exercise.

**Table 2 healthcare-11-02914-t002:** Demographic and clinical profiles of the study participants.

Variables	CGH Group (*n* = 16)	DHC Group (*n* = 16)	SE Group (*n* = 16)	*p*-Value
Height (cm) *	174.00 ± 8.29	168.75 ± 7.72	172.69 ± 10.62	0.353 ^†^
Age (year) *	35.66 ± 4.37	35.72 ± 5.65	37.08 ± 4.92	0.742 ^†^
Weight (kg) *	72.25 ± 13.49	64.41 ± 15.57	68.47 ± 13.66	0.383 ^†^
Constant–Murley score (score) *	53.27 ± 8.00	54.93 ± 8.02	55.86 ± 10.52	0.772 ^†^
Pain (0–15 point)	7.67 ± 2.23	8.25 ± 2.01	8.92 ± 2.11	0.362 ^†^
ADL (0–20 point)	13.00 ± 1.91	13.17 ± 2.41	12.42 ± 2.23	0.682 ^†^
ROM (0–40 point)	27.00 ± 4.55	27.83 ± 2.62	27.83 ± 4.47	0.840 ^†^
Strength (0–25 point)	5.60 ± 5.09	5.68 ± 4.41	6.70 ± 5.86	0.846 ^†^
Rt side/Lt side (%) ^§^	10 (62.50)/6 (37.50)	9 (56.25)/7 (43.75)	9 (56.25)/7 (43.75)	0.918 ^‡^
Female/male (%) ^§^	8 (50.00)/8 (50.00)	9 (56.25)/7 (43.75)	8 (50.00)/8 (50.00)	0.920 ^‡^
Neer test positive (%) ^§^	10 (62.50)	11 (68.75)	10 (62.50)	0.913 ^‡^
Hawkins test positive (%) ^§^	15 (93.75)	14 (87.50)	14 (87.50)	0.800 ^‡^
Yocum test positive (%) ^§^	14 (87.50)	15 (93.75)	14 (87.50)	0.800 ^‡^

* Values are expressed as mean ± SD. ^§^ Values are expressed as numbers (%). ^†^ One-way analysis of variance used for comparison of continuous variables. ^‡^ Chi-squared test used for comparison of categorical variables. CGH, centralization of the glenohumeral joint. DHC, dynamic humeral centering. SE, simple exercise. ADL, activity of daily living. ROM, range of motion. Rt, right. Lt, left.

**Table 3 healthcare-11-02914-t003:** The changes in Constant–Murley score and grip strength.

Parameters	Group	Pre-Treatment	Post-Treatment	Effect Size	*p*-Value
Primary Outcome					
Constant–Murley score (score)	CGH	53.27 ± 8.00	67.62 ± 5.49 ***^††^	2.21	<0.001
	DHC	54.93 ± 8.02	64.25 ± 7.67 ***	1.89	
	SE	55.86 ± 10.52	57.27 ± 10.04	0.28	
Pain (0–15 point)	CGH	7.67 ± 2.23	11.08 ± 1.88 ***	1.82	0.001
	DHC	8.25 ± 2.01	10.92 ± 1.68 ***	1.22	
	SE	8.92 ± 2.11	9.42 ± 1.93	0.96	
ADL (0–20 point)	CGH	13.00 ± 1.91	13.5 ± 1.73	0.40	0.127
	DHC	13.17 ± 2.41	13.08 ± 2.43	0.29	
	SE	12.42 ± 2.23	12.42 ± 2.23	<0.001	
ROM (0–40 point)	CGH	27.00 ± 4.55	35.00 ± 3.95 ***^††^	1.91	<0.001
	DHC	27.83 ± 2.62	33.83 ± 3.66 ***	1.82	
	SE	27.83 ± 4.47	29.83 ± 4.55 *	0.89	
Shoulder Strength (0–25 point)	CGH	5.60 ± 5.09	8.03 ± 4.92 ***	1.21	0.003
	DHC	5.68 ± 4.41	6.42 ± 4.62	0.69	
	SE	6.70 ± 5.86	6.93 ± 6.57	0.18	
Secondary Outcome					
Grip Strength (lb)	CGH	50.63 ± 14.95	57.90 ± 17.80 ***	1.66	<0.001
	DHC	49.74 ± 14.58	50.92 ± 14.61	0.19	
	SE	56.63 ± 17.36	55.22 ± 16.96	0.36	

Values are expressed as the difference between means. The significant difference between groups by repeated measures analysis of variance. CGH, centralization of the glenohumeral joint group. DHC, dynamic humeral centering group. SE, simple exercise group. ADL, activity of daily living. ROM, range of motion. * Significant difference between pre and post treatment, * = *p* < 0.05, *** = *p* < 0.001. ^††^ = *p* < 0.01.

## Data Availability

The data presented in this study are available on request from the corresponding author.

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
