# Peer review of "Short-Term Effects of Centralization of the Glenohumeral Joint and Dynamic Humeral Centering on Shoulder Pain, Disability, and Grip Strength in Patients with Secondary Subacromial Impingement Syndrome"

_healthcare, 2023, doi:10.3390/healthcare11222914_

Round 1

Reviewer 1 Report

Comments and Suggestions for Authors

Thank you for your investigation.

The relationship between the position of the humeral head and therapeutic exercise to control glenohumeral dysfunction is relevant and interesting.

However, the study design and short-term follow-up limit the validity of the study and its impact. The following recommendations are intended to improve your work:

The development of the pathology and its characteristics in the introduction is correct. But it is advisable to better explain the basis of the proposed exercise modalities.

The methodology is good (the inclusion criteria section stands out), but the explanation of the exercises is poor (images should be included). 

The results are limited with a single post-treatment assessment and in the very short term. Why not design a longer intervention and follow-up accordingly?

Author Response

Dear Reviewers,

I would like to thank you for providing the opportunity to revise and resubmit the attached manuscript entitled “Short-term effects of centralization of the glenohumeral joint and dynamic humeral centering on shoulder pain, disability, and grip strength in patients with secondary subacromial impingement syndrome” for publication in Healthcare.

We deeply appreciate the editorial comments and reviewers’ helpful comments on our manuscript which we ignored. We agreed with the points addressed by the Reviewers. We provide our responses to the Reviewers’ comments. Please review the attached files.

Reviewer 2 Report

Comments and Suggestions for Authors

Thank you for the opportunity to review this paper. Although it seems that the authors have tried to provide all relevant information to strengthen their arguments, I think that there are areas that need further improvement. 

Initially, it would be useful to estimate the minimum number of participants needed for this study. This would show the readers that the study has sufficient power.

Then it would be useful to explain why you are using grip strength to examine the effectiveness of your interventions that aim to help the should. Although it can be associated, I do not think that this should be the primary outcome.

Visual aids are missing entirely from the manuscript. 

A thorough analysis of your techniques would also be useful in your case. 

Paragraphs 2.3.1 and 2.3.2 need more analysis to help readers understand what you actually did or instructed patients to do.  

Comments on the Quality of English Language

Moderate editing of English language required

Author Response

(The authors gave the same response as above.)

Reviewer 3 Report

Comments and Suggestions for Authors

Although this could have been an interesting study, as it examines the relative effectiveness of CGH vs. DHC exercises, it has limited clinical generalizability. The design is well thought out (as far as the outcome measures and the content of the exercises included in each group). However, I am not sure why the authors reported the clinical benefit of a single treatment only in patients with SIS.

Line 42: What do you mean by the word “plexus” ?

Final conclusions require expanding.

The effect of the same exercise programs should be presented after 8-10 sessions. A longer term follow-up is also required.

Comments on the Quality of English Language

Adequate

Author Response

(The authors gave the same response as above.)

Round 2

Reviewer 1 Report

Comments and Suggestions for Authors

Thank you for updating your manuscript.

I think the adaptation has been good, and improves the article greatly. The explanation of the exercise modalities is correct. 

I consider that the publication may be relevant but the impact could be limited.

Author Response

We are deeply grateful for your favorable review.
First of all, in order to improve the quality of the main manuscript, we asked native professors working at the school to review it and reflect their comments in the revision.  
You evaluated the research design as [must be improved]. This study clearly followed the clinical research RCT. If the single trial is the problem, it is difficult for us to solve, but if you can understand the value of the single trial a little, we will try to solve it in the subsequent multiple-session study.
We hope that you've satisfied our revisions.

Reviewer 2 Report

Comments and Suggestions for Authors

Accept 

Author Response

All authors are deeply grateful for the reviewer's acceptance.

Reviewer 3 Report

Comments and Suggestions for Authors

The manuscript now sufficiently highlights the limitations pointed out, therefore can proceed to publication, upon Editor's approval.

Comments on the Quality of English Language

Good

Author Response

We are deeply grateful for your favorable review.
You also rated it as [Minor editing of English language required], so we asked a native professor at the school to review it and incorporated their comments into the revision.  
We hope that you've satisfied our revisions.